# Differential Diagnosis of Malignant Lymphadenopathy Using Flow Cytometry on Fine Needle Aspirate: Report on 269 Cases

**DOI:** 10.3390/jcm9010283

**Published:** 2020-01-20

**Authors:** Carla Griesel, Minodora Desmirean, Tonya Esterhuizen, Sergiu Pasca, Bobe Petrushev, Cristina Selicean, Andrei Roman, Bogdan Fetica, Patric Teodorescu, Carmen Swanepoel, Ciprian Tomuleasa, Ravnit Grewal

**Affiliations:** 1National Health Laboratory Services, Tygerberg Hospital, Cape Town 7505, South Africa; cmgriesel84@gmail.com (C.G.); testerguizen@yahoo.com (T.E.); carmens@sun.ac.za (C.S.); ravnitgrewal@gmail.com (R.G.); 2Department of Hematology, Iuliu Hatieganu University of Medicine and Pharmacy, 400124 Cluj Napoca, Romania; dr.minodora.desmirean@gmail.com (M.D.);; 3Department of Pathology, Constantin Papilian Military Hospital, 400001 Cluj Napoca, Romania; 4Research Center for Functional Genomics and Translational Medicine, Iuliu Hatieganu University of Medicine and Pharmacy, 400337 Cluj Napoca, Romania; pasca.sergiu123@gmail.com; 5Department of Pathology, Octavian Fodor Regional Institute for Gastroenterology, 400111 Cluj Napoca, Romania; bobe.petrushev@gmail.com; 6Department of Hematology, Ion Chiricuta Clinical Cancer Center, 400015 Cluj Napoca, Romania; 7Department of Radiology, Iuliu Hatieganu University of Medicine and Pharmacy, 400124 Cluj Napoca, Romania; 8Faculty of Natural Sciences, University of Western Cape, Belville 7535, South Africa; 9The South African National Bioinformatics Institute, Medical Research Council, University of the Western Cape, Belville 7535, South Africa

**Keywords:** fine needle aspiration, flow cytometry, lymphoma

## Abstract

Introduction: Fine needle aspiration (FNA) is frequently the first noninvasive test used for the diagnostic workup of lymphadenopathy. There have been many studies showing its usefulness, especially in conjunction with other techniques for the diagnosis of lymphoma, but it remains inferior to histological examination. The data regarding this subject have mostly been reported mostly from first-world countries, but are scarce for emerging economies. Thus, the current study assesses the agreement between fine needle aspiration flow cytometry (FNA FC) and histology in the aforementioned region. Material and Methods: We conducted a retrospective study including the FNA FC adenopathy diagnoses made between January 2011 and December 2016 at the Tygerberg Hospital, Cape Town, South Africa. Additional variables included were the histological diagnosis, sex and age of the included patients. Results: In the descriptive part of the current study, 269 FNA FC samples were included. The most frequent diagnoses made on these were represented by B-cell lymphoma, reactive adenopathy, no abnormality detected (NAD), and non-hematological malignancy. In the analytical part of the current study, there were 115 cases included that had both valid FNA FC and histological diagnoses. It could be observed that FNA FC can correctly diagnose B-cell lymphoma in most cases, but it is a poor diagnostic tool especially for Hodgkin lymphoma in this setting as only a four-color flow cytometer was available for diagnosis. Moreover, FNA FC diagnosis of reactive adenopathy and of no abnormalities detected was shown to frequently hide a malignant disease. Conclusion: In countries with scarce resources, FNA FC represents a useful diagnostic tool in the case of B-cell lymphoma, but may misdiagnose reactive adenopathy. Thus, FNA FC should be used in a case-specific manner, in addition to as a screening tool, with the knowledge that in cases with a high clinical suspicion of lymphoma, histological diagnosis is a necessity.

## 1. Introduction

Fine needle aspiration (FNA) is often one of the first noninvasive investigations employed when a patient presents with lymphadenopathy. It is a relatively painless, fast, simple, and safe procedure which can be performed in an outpatient setting by a trained professional. In the past ten years, numerous studies have reported FNA to have high diagnostic sensitivity and specificity for lymphoid malignancies when combined with supplementary techniques [1]. Controversy regarding the accuracy of a diagnosis based on FNA alone also exists as FNA has been proven to occasionally give false negative results [2]. At first, this method was used for the diagnosis of recurrent disease and for staging purposes, but its utility in the primary diagnosis of lymphoid malignancies is controversial [3]. This arises from the fact that architecture is an essential component for the diagnosis of some lymphomas, a component that is lost in FNA [2]. Additionally, the World Health Organization (WHO) 2016 guidelines still regard histology of a lymph node as the gold standard for the diagnosis of lymphoma [1]. However, with the advent of techniques such as flow cytometry (FC) and molecular biology assays, there is a debate as to whether FNA can, at least in some part, replace histology, as the latter presents its own limitations. These include analytical subjectivity of the observer, limited reproducibility, a lack of consensus in quantifying antigen expression and higher invasiveness compared to FNA, which requires more extensive preparation [4,5]. In addition, the evaluation of histological samples is also time consuming, which could affect patient management. In view of the high sensitivity achieved with these new techniques, there is speculation that in future, histology may not be critical for the diagnosis of lymphoma. FNA FC is a fast diagnostic method to identify lymphomas, but it has to be combined with complementary methods and clinical presentation in order to arrive at the most accurate diagnosis [4,6]. Some centers have already replaced histology with FNA cytology and complementary methods for the diagnosis of lymphoma relapse [7]. 

Currently, FNA FC is not considered a feasible diagnostic procedure for adenopathy, but represents an initial step in the diagnostic procedure.

While there is much data available regarding the utility of FNA FC worldwide, there is a paucity of similar data in the South African setting. Thus, the present study focused on evaluating the diagnostic utility of FNA FC at Tygerberg hospital (TBH), Cape Town, South Africa.

## 2. Material and Methods

### 2.1. Patient Selection

In the current study, we included patients that had FNA FC for lymphadenopathy at the Tygerberg hospital (TBH), South Africa, between January 2011 and December 2016. The FNA FC and histological diagnoses represented the main variables included. Additionally, we included the patient’s sex and age.

The study is in agreement with the declaration of Helsinki and was approved by the Health Research Ethics Committee of the Faculty of Medicine and Health Sciences of the University of Stellenbosch (Ref nr: S17/07/121), as well as of the Iuliu Hatieganu University of Medicine and Pharmacy/Ion Chiricuta Clinical Cancer Center in Cluj Napoca.

Before the FNA procedure, clinical information was communicated to the patient regarding the process of the procedure itself, as well as its potential risks and benefits. Informed consent was obtained. To perform an FNA, the mass or lymph node was identified and palpated by the pathologist or visualized using ultrasound imaging by an interventional radiologist. Once the mass was identified, the skin over the mass was cleaned with alcohol. A small (23–27 gauge) needle with or without an attached syringe was inserted into the targeted lesion. While employing suction, a brisk back-and-forth motion was applied to the needle, and a sample was collected. The flow cytometry findings included a description of any atypical or clonal lymphoid populations, and a report was generated for the cytopathologist to incorporate into the final cytology diagnosis. For each case, standard panels were applied. For example, when blasts are noted, the acute panel is used. When lymphocytes with plasmacytoid morphology are noted, the plasma cell panel may be incorporated in the chronic panel. This rules out the possibility of a wrong choice of antibody as a cause of wrong diagnosis.

### 2.2. Flow Cytometry Sample Preparation

FNA needle rinses were washed twice with phosphate-buffered solution (PBS), spun at 800× *g* for 3 min and the supernatant removed. Washed samples were then diluted to an appropriate cell concentration using HAMS solution. A 100 μL aliquot of the cell sus- pension was incubated for 15 min in the dark with 20 μL of the appropriate antibody cocktail from the panels listed in Table 1. After incubation, red blood cells were lysed with 2 mL of Becton Dickinson (BD) FACS™ lysing solution for 8 min. At this point, 50 μL of DRAQ5 was added to tube 4 so that nucleated cells could be clearly distinguished by the flow cytometer. All tubes were then washed with PBS, centrifuged at 800× *g* for 3 min, the supernatant removed, and the cells fixed with 250 μL of 1% para-formaldehyde in PBS.

### 2.3. Immunophenotyping

Following collection, the FNA sample was immediately sent to the FC laboratory. The white cell count (WCC) was measured following analysis on a hematology analyzer (ADVIA 2120). A cytospin slide was prepared and stained with May–Grünwald–Giemsa (MGG) (Merck, Darmstadt, Germany) for assessment by the hematopathologist for the decision on the antibody panel.

Samples were subsequently analyzed on the FACSCalibur (Becton Dickinson, CA, USA) for four-color immunophenotyping. Kaluza Analysis version 1.3 (Beckman Coulter, Life sciences, South Africa) and BD CellQuest Pro version 5.2.1. (Becton Dickinson Immunocytometry systems, San Jose, CA, USA) were used for gating and subsequent analysis (Becton Dickinson Immunocytometry systems, San Jose, CA, USA). The gating strategy for data analysis was determined using the leucocyte common antigen, Cluster of differentiation (CD45) positive vs. Side Scatter pattern. Acquisition was set at 10000 CD45-positive events (leucocytes) per tube. Dot plot evaluation of antigen expression was used to determine the cell populations of interest. Clonal populations, if present, were identified and immunophenotyped. If aberrant T-cell populations or plasma cell populations were suspected, further investigation with appropriate extended panels was performed. We made use of standard panels (Table 1) that were applied in the case of suspected, for example, acute leukemia. Occasional extra antibodies will be added according to the discretion of the pathologist involved.

### 2.4. Histology

Tissue samples were fixed in 10% formalin (Merck, South Africa) and subsequently dehydrated with alcohol (Purple Moss, South Africa). Once the alcohol was removed and replaced with xylene (Kimix, South Africa), the sample was impregnated with paraffin wax and embedded in a paraffin block. The samples were then cut to a thickness of 3–4 μm and stained with hematoxylin and eosin (H&E). Expert anatomical pathologists assessed and reported on each case. Immunohistochemistry was ordered as per the pathologist’s discretion.

### 2.5. Data Analysis

Data analysis was performed using R3.5.3. Categorical data was represented as absolute value (percent). Contingency tables were assessed using Fisher’s test. If Fisher’s test was statistically significant for contingency tables with more than two levels in at least one of the variables, we further performed multiple comparison Fisher’s test with Benjamini–Hochberg correction. Normality of the distribution was assessed using Shapiro test and histogram visualization, but also took into consideration the sample size. Non-normally distributed continuous variables were represented as median (quartile 1, quartile 3). Differences between more than two non-normally distributed variables were assessed using Kruskal–Wallis test. If Kruskal–Wallis was statistically significant, a multiple comparison Wilcox test with Benjamini–Hochberg correction was performed. To determine the predictability of a model, we performed a random forest with 2000 trees and a visual assessment of receiver operating characteristic (ROC) curves and their area under the curve (AUC). A *p* value under 0.05 was considered to indicated statistical significance. The ROC curves had a dichotomic variable as an input (0, 1), which is why they have an odd appearance. Table 2 presents the agreement between the FC diagnosis and the histological diagnosis, which offers a better representation.

## 3. Results

In the descriptive part of the current study, 269 patients with FNA FC samples were included. Of these, 122 (45.4%) were male and 147 (54.6%) were female. The median age was 42 (29,58) years.

In Figure 1, we represented the distribution of the diagnoses by FNA FC and the number of those that had subsequent histological evaluation. In the analytical part of the current study, we included only the cases in which both FNA FC and histology were valid, resulting in the inclusion of 115 samples.

We performed an exploratory analysis by generating a chord diagram between the FC FNA diagnosis and the subsequent histological diagnosis (Figure 2). The most frequent diagnoses made by FNA FC were represented by B-cell lymphoma, reactive, no abnormalities detected (NAD), and non-hematological malignancies. Further, we generated ROC curves to assess the predictability that FNA FC can have for histology (Figure 3). Redundant representations were not shown. Hodgkin lymphoma was not included because there was no reported diagnosis of Hodgkin lymphoma through FNA FC. Chronic lymphocytic leukemia (CLL) and natural killer (NK) cell lymphoma were not included because all the cases diagnosed through FNA FC perfectly matched the histological diagnosis.

In Figure 4, we represented the age and sex distribution of patients with different histological diagnoses in our cohort. The age difference between the multiple histology diagnoses was statistically significant (*p* = 0.00045). In the pairwise Wilcox test, the only significant difference in age was represented by the younger age in non-hematological malignancies compared to B-cell lymphoma. Nonetheless, it must be noted that the age distribution of patients with Hodgkin lymphoma and reactive adenopathy also approached statistical significance when compared to B-cell lymphoma patients. Statistical significance was also reached when analyzing the contingency table between sex and histological diagnosis (*p* = 0.0081). In the multiple comparison Fisher’s test, there was no statistically significant result, but there was a tendency for patients with Hodgkin lymphoma to be more frequently males, while B-cell lymphoma and reactive adenopathy patients were more frequently women.

When using a random forest algorithm for predicting histology diagnosis from FNA FC diagnosis, most diagnoses did not have an acceptable prediction rate, with the best being represented by B-cell lymphoma with an error rate of 10.41%. When also including sex and age as input variables, the error rate for B-cell lymphoma dropped to 6.25% with the rest of the diagnoses still not reaching a clinically feasible error rate. It has to be mentioned that the first performed random forest had the role of acting as a baseline for comparison with the latter random forest.

## 4. Discussion

Hematological malignancies such as leukemias and lymphomas whilst rare in the past, are becoming increasingly diagnosed, especially B cell lymphomas and usually present with lymphadenopathy as a prominent feature. Thus, pathological sampling with minimally invasive modalities, such as fine needle aspiration (FNA) or an excisional biopsy, is typically performed when there is persistent lymphadenopathy, clinical symptoms, or radiological imaging findings indicating potential malignancy. Although excisional biopsy of a lymph node provides a large amount of tissue, there are risks associated with this surgical procedure, including the risks associated with anesthesia and the risks of a surgical biopsy in the case of bleeding, infection, and nerve damage. Thus, FNA provides a less invasive, quick and cost-effective modality for tissue sampling and triage of these lesions for ancillary studies like flow cytometry (FC), which can help to further characterize a patient’s lymphadenopathy especially in countries where Tuberculosis (TB) and Human Immunodeficiency Viral (HIV) are prevalent. FNA has been found to be a sensitive and specific diagnostic tool for differentiating between benign and malignant lymphadenopathies [8,9]. When coupled with FC, FNA can be an even more powerful diagnostic tool because hematopoietic malignancies can more confidently be either diagnosed or excluded, and further subclassification of lymphoid populations can be provided. Combining FNA with FC has been previously shown to be an accurate tool for distinguishing benign processes from malignant ones.

Although reactive lymph node hyperplasia is the most common cause of lymphadenopathy, the persistence of enlarged lymph nodes can often worry patients, which can lead to further investigation. Thus, young patients with persistent lymphadenopathy are frequently referred for a biopsy, given that clinicians do not want to miss a case of malignancy in a patient, and since watchful waiting may lead to a delayed diagnosis. All cases of non-hematological diagnoses were reactive lymphadenopathy and no further investigations were required.

The current study has shown that in our center’s experience, most B-cell lymphomas can be reliably diagnosed using FNA FC even when using a four-color flow cytometer. As a proof-of-concept, we present the dot blot-based gating strategy of a small lymphocytic lymphoma case in Figure 5A–L. As mentioned before, FNA FC can be useful to detect lymphoma recurrence, but there is still a question of whether it is feasible to be used in the primary diagnosis of lymphoma [6,10,11]. This principle has reached the point in which there are authors that consider FNA FC sufficient to detect head and neck lymphomas without the need for subsequent histological examination on surgical biopsy [12]. Moreover, FNA FC showed its utility in a deep-seated lymphoma setting where surgical biopsy would present a higher degree of difficulty and risk [13,14,15]. In the case of follicular lymphoma, it has been shown that FNA FC can diagnose most cases with the exception of some variants that can mimic diffuse large B-cell lymphoma [16]. Nonetheless, these results cannot be generalized in the case of T-cell lymphomas and Hodgkin lymphomas as FNA FC diagnostic accuracy in these cases is still lacking, especially when using a four-color flow cytometer [17].

It should be mentioned that in the current study, no Hodgkin lymphomas were diagnosed by FNA FC although they were present for histological diagnosis. As has been shown by others, Hodgkin lymphoma represents a common error source [17,18,19]. In the current study, most cases of Hodgkin lymphoma were seen as reactive or normal, and only a few as B-cell lymphoma on FNA FC, which is probably due to the disease characteristics. More specifically, because the Reed–Sternberg cells were large and represented only approximately 5% of the tumor mass, the rest were represented by a reactive infiltrate [20].

Others have shown that in benign adenopathy, the immunophenotype detected by FNA FC is similar to the one detected by histology [21]. Nonetheless, the aforementioned study included only histologically confirmed benign adenopathy and, thus, the common malignant confounders were not present.

We have also shown that FNA FC correctly identified all CLL samples but, because there were only three patients with CLL that had both FNA FC and histology, we cannot conclude that FNA FC is a perfect method for detecting CLL adenopathy. Nonetheless, FC represents a must in the diagnosis workup of CLL, so we would expect that this method is accurate for this diagnosis. Moreover, the clinical diagnosis of CLL is generally made on peripheral blood and/or on bone marrow aspirate, so CLL diagnosed from an extramedullary mass is generally rare [22,23,24,25].

The other entity correctly diagnosed by FNA FC was represented by the NK cell lymphoma, but this was also represented by a limited sample of one. Nonetheless, although NK cell lymphomas represent rare entities, they can be relatively easily diagnosed through FNA FC because of their specific immunophenotype [26,27].

Other factors related to the methodological process or the adenopathy architecture can also constitute error sources such as sample contamination or uneven involvement of the lymph node [28].

Although FNA FC represents a relatively reliable diagnosis tool for B-cell lymphomas that can be used in a broad range of centers, more advanced complementary methods, specifically molecular and imaging studies, have been shown to greatly enhance the diagnostic accuracy of FNA in other fields also, thus moving to a less invasive and quicker diagnostic method for adenopathy [29,30,31].

Moving away from the clinical implications, it also must be considered that formalin-fixed paraffin embedded (FFPE) samples can be stored for longer, therefore ensuring that retrospective studies can be conducted when compared to flow cytometry. Thus, whilst histology remains the recommended tool for accurate diagnosis of lymphoma, flow cytometry does aid in quick assessments, especially when the clinical presentation requires immediate results [32,33,34,35,36,37,38,39,40]. 

Nonetheless, one caveat that can also be encountered is in the case of a small aspirate sample. In these cases, FISH or sequencing assessments of the cell population can indicate the diagnosis in addition to also having a prognostic value. These changes can be represented by Tumor protein p53 TP53 and cyclin-dependent kinase Inhibitor 2A CDKN2A mutations as markers of transformation [41], or can assist in differentiating between activated peripheral B-cell (ABC) and germinal center B-cell (GCB) types of DLBCL [42]. Moreover, certain translocations can be identified by FISH with the additional value of providing orientation in the diagnosis of a certain type of lymphoma [43].

## 5. Conclusions

FNA FC using four-color flow cytometry, as was the case in this study, represents a useful diagnostic tool for the diagnosis of B-cell lymphomas. However, it is difficult to completely exclude reactive adenopathy, especially where the index of suspicion for lymphoma clinically was high, as well as to diagnose Hodgkin lymphoma or T-cell lymphoma. Thus, in low-to-middle income countries, it is a good ancillary tool to aid in management decisions of B-cell lymphomas, especially when clinical decisions need to be made immediately, and when access to the lymphoid site affected was almost impossible. 

## Figures and Tables

**Figure 1 jcm-09-00283-f001:**
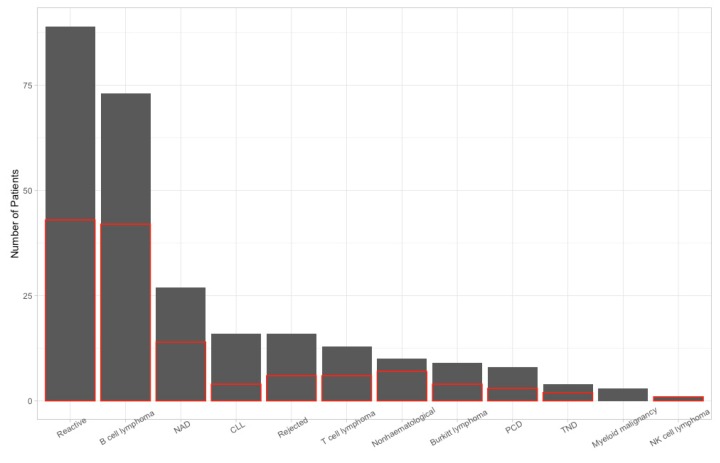
Bar plot representing the absolute number of patients with each fine needle aspiration flow cytometry (FNA FC) diagnosis (grey fill) and how many of these cases subsequently had a histological evaluation (red outline). NAD: no abnormalities detected; CLL: chronic lymphocytic leukemia; PCD: plasma cell dyscrasia; NK cell lymphoma: natural killer cell lymphoma; TND: test not done; Non-hematological: non-hematological malignancy; Reactive: reactive inflammatory infiltrate.

**Figure 2 jcm-09-00283-f002:**
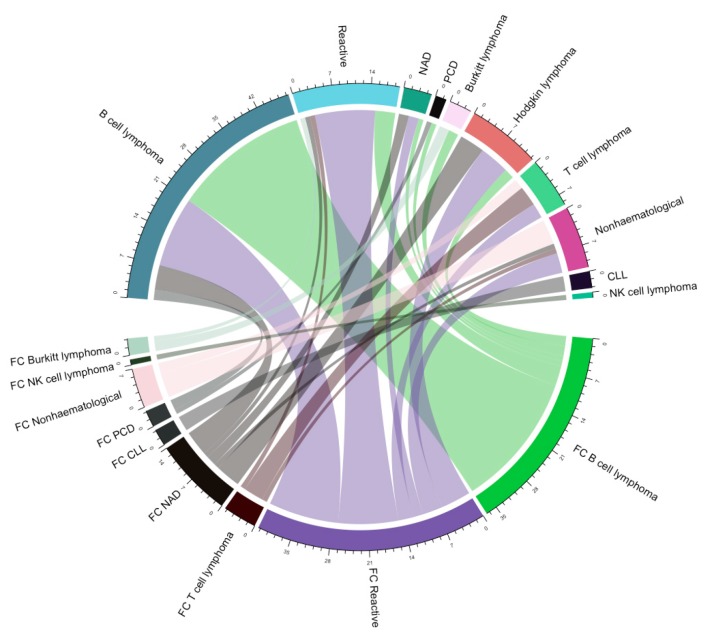
Chord diagram representing the agreement between FNA FC diagnoses (represented with FNA FC diagnoses with FC before their name) and histological diagnoses (represented with histological diagnoses). NAD: no abnormalities detected; CLL: chronic lymphocytic leukemia; PCD: plasma cell dyscrasia; NK cell lymphoma: natural killer cell lymphoma.

**Figure 3 jcm-09-00283-f003:**
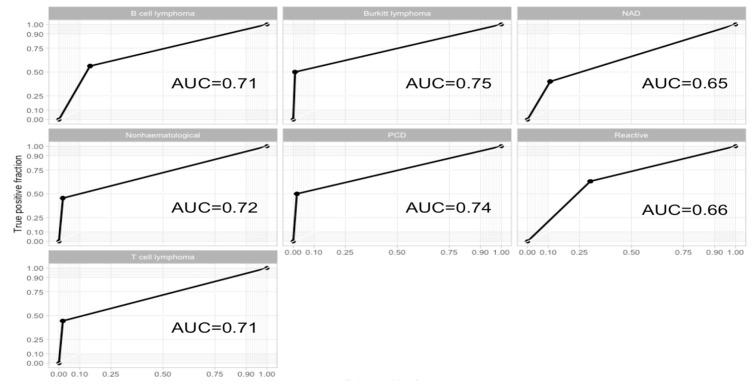
ROC curves and AUC assessing the agreement between FNA FC and histology. NAD: no abnormalities detected; PCD: plasma cell dyscrasia; NK cell lymphoma: natural killer cell lymphoma; Non-hematological: non-hematological malignancy. ROC: receiver operating characteristics. AUC: Area under the Curve.

**Figure 4 jcm-09-00283-f004:**
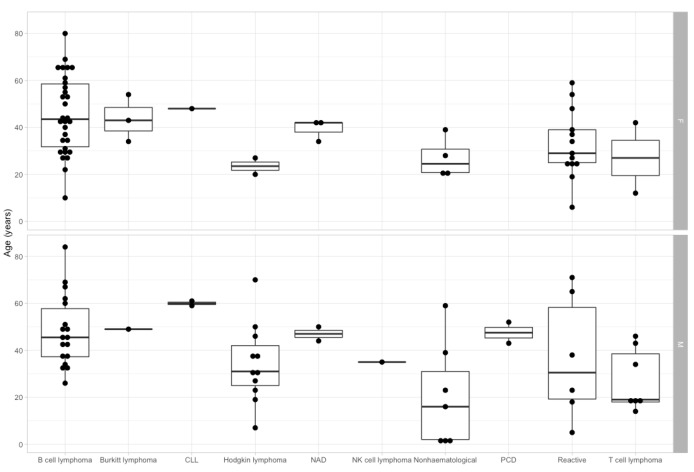
Patient’s age and sex distribution between different histological diagnoses. PCD: plasma cell dyscrasia; NK cell lymphoma: natural killer cell lymphoma; Non-hematological: non-hematological malignancy. CLL – chronic lymphocytic leukemia. NAD – No abnormality detected.

**Figure 5 jcm-09-00283-f005:**
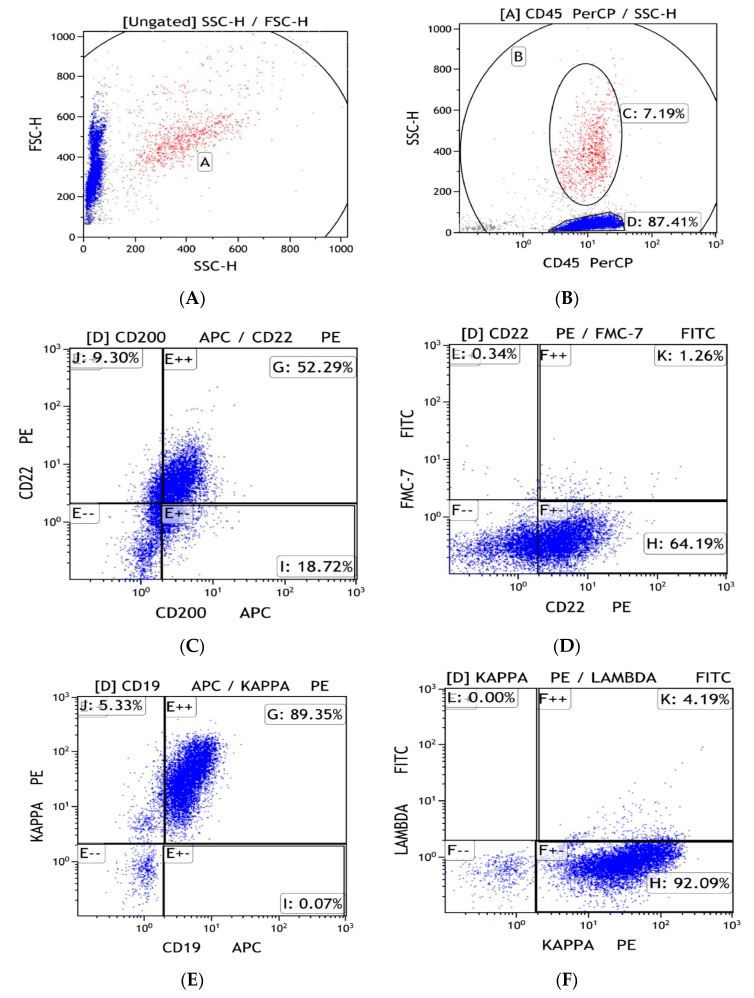
(**A**–**L**). Dot blot-based gating strategy for a proof-of-concept study of small lymphocytic leukemia.

**Table 1 jcm-09-00283-t001:** Four-color panels column 1: tubes containing the four monoclonal antibodies conjugated to the respective fluorescent dyes. Column 2, 3, 4, 5: represent the fluorescent dyes.

*1.*	*2*. FITC	*3*. PE	*4*. PerCP	*5*. APC
**Chronic panel**				
1	CD8	CD4	CD45	CD3
2	CD5	CD23	CD45	CD19
3	CD20	CD10	CD45	CD38
4	FMC-7	CD22	CD45	CD200
5	Lambda	Kappa	CD45	CD19
6	CD10	CD34	CD45	CD19
**Plasma cell panel**				
1	CD8	CD4	CD45	CD3
2	CD20	CD79a	CD45	CD38
3	CD56	CD138	CD45	CD38
4	CD56	CD10	CD45	CD38
5	cLambda	cKappa	CD45	CD38
**Chronic T-cell panel**				
1	CD8	CD4	CD45	CD3
2	CD5	CD23	CD45	CD19
3	CD20	CD10	CD45	CD38
4	Lambda	Kappa	CD45	CD19
5	CD7	CD1a	CD45	CD2
6	CD25	CD4	CD45	CD2
7	CD16	CD30	CD45	
8	CD56	CD10	CD45	CD38
9	CD57	CD8	CD45	CD3
Cytoplasmic Markers				
10		cCD79a	CD45	cCD3
**Acute Leukemia panel**				
1	CD8	CD4	CD45	CD3
2	CD10	CD34	CD45	CD19
3	HLADR	CD33	CD45	CD11b
4	CD7	CD34	CD45	CD2
5	CD56	CD13	CD45	CD11b
6	CD15	CD117	CD45	
7	CD14	CD64	CD45	
Cytoplasmic Markers				
1	cMPO	cCD79a	mCD45	cCD3
2		cIgM	mCD45	mCD19
3	cTdT control		mCD45	mCD19
4	cTdT Test	cCD22	mCD45	mCD19

FITC: fluorescein isothiocyanate, PE: phycoerythrin, PerCP: peridinin chlorophyll protein, APC: allophycocyanin, TdT: terminal deoxynucleotidyl transferase.

**Table 2 jcm-09-00283-t002:** Comparison between flow cytometry and histology diagnosis.

Flow Cytometry Diagnosis	Agreement with Histology
B-cell lymphoma	73%
Burkitt lymphoma	66.7%
CLL	100%
NAD	14.3%
NK cell lymphoma	100%
Non-hematological	71.4%
PCD	33.3%
Reactive	29.3%
T-cell lymphoma	66.7%

CLL: chronic lymphocytic leukemia; NAD: no abnormality detected; NK: natural killer; PCD: plasma cell dyscrasia.

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
