# Peer review of "Differential Diagnosis of Malignant Lymphadenopathy Using Flow Cytometry on Fine Needle Aspirate: Report on 269 Cases"

_jcm, 2020, doi:10.3390/jcm9010283_

Round 1
Reviewer 1 Report
This article is an interesting study demontrating the usefulness of fine needle aspiration flow cytometry for B cell lymphoma diagnosis.
This paper is very interesting but many important informations are missing.
First :
In patient selection part : HIV status need to be added for all patients or at leat for the 115 patients with FNA FC and histology.
Immunophenotyping part:
a table with all the antibodies used must be added (with company, cat number and dilution used if accurate)
Please explain how the antibody panel is chosen after examination by the hematopathologist? detail all antibodies used in each case of suspicion of one disease: ex in case of CLL suspicion, the antibody panl comprised anti-CD20, CD5...
A discussion about this choice of antibodies need to be added as well: maybe the choice of antibody was not accurate and that is why a wrong diagnosis was done?
In results part:
The authors should detail all the non-haematological diagnoses performed.
PLease explain the sentence p 3 l132 to 133 as it is not clear: "Statistcal significance was also reached ....". what did the authors compare?
In discussion, please discuss the molecular biology of lymphoma as an additional help for lymphoma diagnosis in small samples
Author Response
Reviewer #1:
- A table with all the antibodies used must be added. (with company, cat number and dilution used if accurate).
Thank you very much for an important feed-back. We have added in the revised manuscript that:
“Flow cytometry sample preparation.
FNA needle rinses were washed twice with phosphate buffered solution (PBS), spined at 800 g for 3 minutes and the supernatant removed. Washed samples were then diluted to an appropriate cell concentration using HAMS solution. 100 μL of the cell sus‐ pension was incubated for 15 minutes in the dark with 20 μL of the appropriate antibody cocktail from the panels listed in Table 1. After incubation, red blood cells were lysed with 2 mL of Becton Dickinson (BD) FACS™ lysing solution for 8 minutes. At this point 50 μL of DRAQ5 was added to tube 4 so that nucleated cells could be clearly distinguished by the flow cytometer. All tubes were then washed with PBS, centrifuged at 800 g for 3 minutes, the supernatant removed, and the cells fixed with 250 μL of 1% para‐ formaldehyde in PBS. “, as well as “The gating strategy for data analysis was determined using the leucocyte common antigen (CD45) positive vs Side Scatter pattern. Acquisition was set at 10 000 CD45 positive events (leucocytes) per tube. Dot plot evaluation of antigen expression was used to determine the cell populations of interest. Clonal populations, if present, were identified and immunophenotyped. If aberrant T cell populations or plasma cell populations were suspected, further investigation with appropriate extended panels was performed. “
Please explain how the antibody panel is chosen after examination by the hematopathologist? detail all antibodies used in each case of suspicion of one disease: ex in case of CLL suspicion, the antibody panel comprised anti-CD20, CD5.
Thank you very much for an important feed-back. We have added in the revised manuscript that “The gating strategy for data analysis was determined using the leucocyte common antigen (CD45) positive vs Side Scatter pattern. Acquisition was set at 10 000 CD45 positive events (leucocytes) per tube. Dot plot evaluation of antigen expression was used to determine the cell populations of interest. Clonal populations, if present, were identified and immunophenotyped. If aberrant T cell populations or plasma cell populations were suspected, further investigation with appropriate extended panels was performed. We make use of standard panels (Table 1) that is applied in the case of suspicion of, for example, acute leukaemia. Occasional extra antibodies will be added according to the discretion of the pathologist involved.“
A discussion about this choice of antibodies need to be added as well: maybe the choice of antibody was not accurate and that is why a wrong diagnosis was done?
Thank you very much for an important feed-back. We have added in the revised manuscript that “Before the FNA procedure, clinical information was communicated to the patient regarding the process of the procedure itself, as well as its potential risks and benefits. Informed consent was obtained. To perform an FNA, the mass or lymph node was identified and palpated by the pathologist or visualized using ultrasound imaging by an interventional radiologist. Once the mass was identified, the skin over the mass was cleaned with alcohol. A small (23-27 gauge) needle with or without an attached syringe was inserted into the targeted lesion. While employing suction, a brisk back-and-forth motion was applied to the needle, and a sample was collected. The flow cytometry findings included a description of any atypical or clonal lymphoid populations, and a report was generated for the cytopathologist to incorporate into the final cytology diagnosis. For each case, standard panels are applied. For example, when blasts are noted, the acute panel is used. When lymphocytes with plasmacytoid morphology is noted, the plasma cell panel may be incorporated in the chronic panel. These rules out the possibility of a wrong choice of antibody as a cause of wrong diagnosis. “.
The authors should detail all the non-haematological diagnoses performed.
Thank you very much for an important feed-back. We have added in the revised manuscript that “Hematological malignancies such as leukemias and lymphomas are rare overall, but they do comprise a large percent of malignancies and can present with lymphadenopathy as a prominent feature. Thus, pathological sampling with minimally invasive modalities, such as fine-needle aspiration (FNA), or an excisional biopsy is typically performed when there is persistent lymphadenopathy, clinical symptoms, or radiological imaging findings concerning for malignancy. Although excisional biopsy of a lymph node provides a large amount of tissue, there are risks associated with this surgical procedure, including the risks associated with anesthesia and the risks of a surgical biopsy, as is the case of bleeding, infection and nerve damage. Thus, FNA provides a less invasive, quick, and cost-effective modality for tissue sampling and triage of these lesions for ancillary studies like flow cytometry (FC), which can help to further characterize a patient’s lymphadenopathy. FNA has been found to be a sensitive and specific diagnostic tool for differentiating between benign and malignant lymphadenopathies. When coupled with FC, FNA can be an even more powerful diagnostic tool because hematopoietic malignancies can more confidently be ruled in or out, and further subclassification of lymphoid populations can be provided. Combining FNA with FC has been previously shown to be an accurate tool for distinguishing benign processes from malignant ones.
Although reactive lymph node hyperplasia is the most common cause of lymphadenopathy, the persistence of enlarged lymph nodes can often worry patients, which can lead to further investigation. Thus, young patients with persistent lymphadenopathy are frequently referred for a biopsy, given that clinicians do not want to miss a case of malignancy in a patient and since watchful waiting may lead to a delayed diagnosis. All cases of non-hematological diagnoses were reactive lymphadenopathy and no further investigations were required. “
Please explain the sentence p 3 l132 to 133 as it is not clear: "Statistcal significance was also reached ....". what did the authors compare?
Thank you very much for an important feed-back. We assessed if there was any difference in disease distribution by gender. The contingency table analyzed contained gender and histological diagnosis. This has been done so that we could further screen where did this difference result from.
In discussion, please discuss the molecular biology of lymphoma as an additional help for lymphoma diagnosis in small samples.
Thank you for the very important feed-back. We have attached the following phrases and references that” Nonetheless, one caveat that can also be encountered is in the case of a small aspirate sample. In these cases, FISH or sequencing assessments of the cell population can orient to the diagnosis and while also having a prognostic value. These changes can be represented by TP53 and CDKN2A mutations as markers of transformation [39]; or can orient in differentiating between activated peripheral B-cells (ABC) and germinal center B-cells (GCB) types of DLBCL [40]. Moreover, certain translocations can be identified by FISH with additional value in orienting to the diagnosis of a certain type of lymphoma [41].”

Reviewer 2 Report
Dear Authors,
your study is of value and may be worth publishing, but a lot more details is needed. The main focus of your study is flow cytometry and what you are doing is completely avoiding describing it. You have to provide a lot more details about how the FC was performed. Please refer to: https://www.ncbi.nlm.nih.gov/pmc/articles/PMC2773297/
Please provide a more comprehensive description of methods. It is advisable to follow the minimal guidelines for reporting FC studies. Therefore, please provide details about antibodies used and procedures followed. Without details about how FC was performed, the study has very limited value. Were the Euroflow recommendations followed? Or any other international recommendation for flow cytometry haematological diagnostics? Similarly, please provide more information about IHC - abs, procedure etc. Please explain abreviations in figure captions Please explain how the ROC curves were prepared. Something is wrong with them, that is not how ROC curve should look like.Author Response
Reviewer #2:
Our study is of value and may be worth publishing, but a lot more details is needed. The main focus of your study is flow cytometry and what you are doing is completely avoiding describing it. You have to provide a lot more details about how the FC was performed. Please provide a more comprehensive description of methods. It is advisable to follow the minimal guidelines for reporting FC studies. Therefore, please provide details about antibodies used and procedures followed. Without details about how FC was performed, the study has very limited value. Were the Euroflow recommendations followed? Or any other international recommendation for flow cytometry haematological diagnostics? Similarly, please provide more information about IHC - abs, procedure etc.
Thank you very much for an important feed-back. We have added in the revised manuscript that “Thank you very much for an important feed-back. We have added in the revised manuscript that:
“Flow cytometry sample preparation.
FNA needle rinses were washed twice with phosphate buffered solution (PBS), spined at 800 g for 3 minutes and the supernatant removed. Washed samples were then diluted to an appropriate cell concentration using HAMS solution. 100 μL of the cell sus‐ pension was incubated for 15 minutes in the dark with 20 μL of the appropriate antibody cocktail from the panels listed in Table 1. After incubation, red blood cells were lysed with 2 mL of Becton Dickinson (BD) FACS™ lysing solution for 8 minutes. At this point 50 μL of DRAQ5 was added to tube 4 so that nucleated cells could be clearly distinguished by the flow cytometer. All tubes were then washed with PBS, centrifuged at 800 g for 3 minutes, the supernatant removed, and the cells fixed with 250 μL of 1% para‐ formaldehyde in PBS. “, as well as “The gating strategy for data analysis was determined using the leucocyte common antigen (CD45) positive vs Side Scatter pattern. Acquisition was set at 10 000 CD45 positive events (leucocytes) per tube. Dot plot evaluation of antigen expression was used to determine the cell populations of interest. Clonal populations, if present, were identified and immunophenotyped. If aberrant T cell populations or plasma cell populations were suspected, further investigation with appropriate extended panels was performed. “
We have also added in the revised manuscript that “Hematological malignancies such as leukemias and lymphomas are rare overall, but they do comprise a large percent of malignancies and can present with lymphadenopathy as a prominent feature. Thus, pathological sampling with minimally invasive modalities, such as fine-needle aspiration (FNA), or an excisional biopsy is typically performed when there is persistent lymphadenopathy, clinical symptoms, or radiological imaging findings concerning for malignancy. Although excisional biopsy of a lymph node provides a large amount of tissue, there are risks associated with this surgical procedure, including the risks associated with anesthesia and the risks of a surgical biopsy, as is the case of bleeding, infection and nerve damage. Thus, FNA provides a less invasive, quick, and cost-effective modality for tissue sampling and triage of these lesions for ancillary studies like flow cytometry (FC), which can help to further characterize a patient’s lymphadenopathy. FNA has been found to be a sensitive and specific diagnostic tool for differentiating between benign and malignant lymphadenopathies [27592067, 26091519]. When coupled with FC, FNA can be an even more powerful diagnostic tool because hematopoietic malignancies can more confidently be ruled in or out, and further subclassification of lymphoid populations can be provided. Combining FNA with FC has been previously shown to be an accurate tool for distinguishing benign processes from malignant ones.
Although reactive lymph node hyperplasia is the most common cause of lymphadenopathy, the persistence of enlarged lymph nodes can often worry patients, which can lead to further investigation. Thus, young patients with persistent lymphadenopathy are frequently referred for a biopsy, given that clinicians do not want to miss a case of malignancy in a patient and since watchful waiting may lead to a delayed diagnosis. All cases of non-hematological diagnoses were reactive lymphadenopathy and no further investigations were required. “.
Please explain abbreviations in figure captions.
Thank you very much for an important feed-back. We have added in the revised manuscript:
“Figure 1. Bar plot representing the absolute number of patients with each FNA FC diagnosis (grey fill) and how many of these cases subsequently had a histological evaluation (red outline). NAD: no abnormalities detected; CLL: chronic lymphocytic leukemia; PCD: plasma cell dyscrasia; NK cell lymphoma: natural killer cell lymphoma; TND: test not done; Non-hematological: non-hematological malignancy; Reactive: reactive inflammatory infiltrate.
Figure 2. Chord diagram representing the agreement between FNA FC diagnoses (represented with FNA FC diagnoses with FC before their name) and histological diagnoses (represented with histological diagnoses).FNA: fine needle aspiration; FC: flow cytometry; NAD: no abnormalities detected; CLL: chronic lymphocytic leukemia; PCD: plasma cell dyscrasia; NK cell lymphoma: natural killer cell lymphoma.
Figure 3. ROC curves and AUC assessing the agreement between FNA FC and histology. NAD: no abnormalities detected; PCD: plasma cell dyscrasia; NK cell lymphoma: natural killer cell lymphoma; Non-hematological: non-hematological malignancy.
Figure 4. Patient’s age and sex distribution between different histological diagnoses. PCD: plasma cell dyscrasia; NK cell lymphoma: natural killer cell lymphoma; Non-hematological: non-hematological malignancy. “
Please explain how the ROC curves were prepared. Something is wrong with them, that is not how ROC curve should look like.
Thank you for your important feed-back. We have added in the revised manuscript that “The ROC curves had a dichotomial variable as an input (0, 1), that is why they have an odd appearance. Table 2 presents the agreement between the flow cytometry diagnosis and the histological diagnosis which offers a better representation. “
Table 2.
Flow cytometry diagnosis |
Agreement with histology |
B cell lymphoma |
73% |
Burkitt lymphoma |
66.7% |
CLL |
100% |
NAD |
14.3% |
NK cell lymphoma |
100% |
Non-hematological |
71.4% |
PCD |
33.3% |
Reactive |
29.3% |
T cell lymphoma |
66.7% |

Round 2
Reviewer 2 Report
At least one representative dot-blot-based gating strategy should be presented. It is not sufficient to describe some basics of gating in methods. Please provide the model of flow cytometer used. Authors claim to use only 4-colour FC, but list antibodies that can be easily composed into 7 or 8 colour panel depending on the flow cytometer. Moreover, authors claim to have used CD45 staining to distinguish between lymphocytes and other cells, but no anti-CD45 antibody is listed in Table 1. Is Table 1 copy-pasted or is methodology copy-pasted from other publication? Please provide full details about the panels used - list tubes with antibodies and provide graphical overview of how it was decided which tubes to include for the particular patient. Why do the authors use two different clones of a-CD3 antibody?Author Response
Dear Editor,
Thank you very much for reviewing our manuscript. We appreciate the tremendous effort and time the reviewers devoted to improving our manuscript. We sincerely feel that their thoughtful comments have further strengthened the manuscript. Specific responses to each comment are presented in the Responses to the Reviewers. In the revised manuscript, revisions to the manuscript are indicated in red font. We hope that our responses to the reviewers’ comments and the revisions made to the manuscript satisfy all questions and concerns.
With my best regards,
Ciprian Tomuleasa, M.D.
Department of Hematology,
Iuliu Hatieganu University of Medicine and Pharmacy, Cluj Napoca, Romania.
Comments to the Reviewers
Reviewer #2:
At least one representative dot-blot-based gating strategy should be presented. It is not sufficient to describe some basics of gating in methods. Please provide the model of flow cytometer used. Authors claim to use only 4-colour FC, but list antibodies that can be easily composed into 7 or 8 colour panel depending on the flow cytometer. Moreover, authors claim to have used CD45 staining to distinguish between lymphocytes and other cells, but no anti-CD45 antibody is listed in Table 1. Is Table 1 copy-pasted or is methodology copy-pasted from other publication? Please provide full details about the panels used - list tubes with antibodies and provide graphical overview of how it was decided which tubes to include for the particular patient.
Thank you very much for an important feed-back. First of all, I would like to with both the reviewers, as well as the editors a Happy New Year and may the year 2020 bring only good health and accomplishments of all their endeavors.
We have updated Table 1 and provided the revised Table 1. The newly updated Tables is:
Table 1: Four-colour Panels Column 1: tubes containing the four monoclonal antibodies conjugated to the respective fluorescent dyes. Column 2,3,4,5: represent the fluorescent dyes.
1. |
2. FITC |
3. PE |
4. PerCP |
5. APC |
Chronic panel |
|
|
|
|
1 |
CD8 |
CD4 |
CD45 |
CD3 |
2 |
CD5 |
CD23 |
CD45 |
CD19 |
3 |
CD20 |
CD10 |
CD45 |
CD38 |
4 |
FMC-7 |
CD22 |
CD45 |
CD200 |
5 |
Lambda |
Kappa |
CD45 |
CD19 |
6 |
CD10 |
CD34 |
CD45 |
CD19 |
Plasma cell panel |
|
|
|
|
1 |
CD8 |
CD4 |
CD45 |
CD3 |
2 |
CD20 |
CD79a |
CD45 |
CD38 |
3 |
CD56 |
CD138 |
CD45 |
CD38 |
4 |
CD56 |
CD10 |
CD45 |
CD38 |
5 |
cLambda |
cKappa |
CD45 |
CD38 |
Chronic T cell panel |
|
|
|
|
1 |
CD8 |
CD4 |
CD45 |
CD3 |
2 |
CD5 |
CD23 |
CD45 |
CD19 |
3 |
CD20 |
CD10 |
CD45 |
CD38 |
4 |
Lambda |
Kappa |
CD45 |
CD19 |
5 |
CD7 |
CD1a |
CD45 |
CD2 |
6 |
CD25 |
CD4 |
CD45 |
CD2 |
7 |
CD16 |
CD30 |
CD45 |
|
8 |
CD56 |
CD10 |
CD45 |
CD38 |
9 |
CD57 |
CD8 |
CD45 |
CD3 |
Cytoplasmic Markers |
|
|
|
|
10 |
|
cCD79a |
CD45 |
cCD3 |
Acute Leukaemia panel |
|
|
|
|
1 |
CD8 |
CD4 |
CD45 |
CD3 |
2 |
CD10 |
CD34 |
CD45 |
CD19 |
3 |
HLADR |
CD33 |
CD45 |
CD11b |
4 |
CD7 |
CD34 |
CD45 |
CD2 |
5 |
CD56 |
CD13 |
CD45 |
CD11b |
6 |
CD15 |
CD117 |
CD45 |
|
7 |
CD14 |
CD64 |
CD45 |
|
Cytoplasmic Markers |
|
|
|
|
1 |
cMPO |
cCD79a |
mCD45 |
cCD3 |
2 |
|
cIgM |
mCD45 |
mCD19 |
3 |
cTdT control |
|
mCD45 |
mCD19 |
4 |
cTdT Test |
cCD22 |
mCD45 |
mCD19 |
Abbreviations: FITC: Fluorescein isothiocyanate, PE: Phycoerythrin, PerCP: Peridinin chlorophyll protein, APC: Allophycocyanin.
As for the dot-blot-based gating strategy, we have added in the revised manuscript that “As a proof-of-concept, we present the dot-blot-based gating strategy of a small lymphocytic lymphoma case in Figures 5A-L. “. We have also added the subsequent 12 new images.
